# Predictive Model for Amotivation and Discipline in Physical Education Students Based on Teaching–Learning Styles

**Marta Leyton-Román** [1] , **Jaime José León González-Vélez** [2] , **Marco Batista** [3] and **Ruth Jiménez-Castuera** [4,*]

1 Sports Studies Center, Rey Juan Carlos University, 28933 Madrid, Spain; marta.leyton@urjc.es
2 Department of Education, University of Las Palmas de Gran Canarias, 35001 Las Palmas, Spain; jaime.leon@ulpgc.es
3 Sport, Health and Exercise Research Unit (SHERU), Polytechnic Institute of Castelo Branco, 6000-084 Castelo Branco, Portugal; marco.batista@ipcb.pt
4 Didactic and Behavioral Analysis in Sport Research Group, Faculty of Sports Science, University of Extremadura, 10003 Cáceres, Spain
* Correspondence: ruthji@unex.es; Tel.: +34-676-751-040

**Abstract:** One of the purposes of teachers is to ensure the motivation of the students in their classes and to maintain disciplined behaviours. However, the teaching styles and methodologies used do not always have a positive effect on student's motivation and discipline. This study analysed the relationship between student's perceptions of the controlling behaviours of their physical education teacher, together with amotivation and discipline styles from Self-Determination Theory. The sample comprised 922 students, aged between 14 and 18 years (M = 14.95; SD = 0.98). Students' perceptions of less controlling discipline styles (control of the use of rewards) negatively predicted the thwarting of autonomy need. Conversely, a more controlling discipline style (judging and devaluing) positively predicted the thwarting of autonomy need, and this, positively predicted amotivation, which negatively predicted disciplinary behaviours and positively predicted undisciplined behaviours. Teachers must avoid using controlling behaviours like judging and devaluing, as this reinforces amotivation towards physical education and undisciplined student behaviours. The importance of designing classes where the student has responsibilities to make decisions and to be part of their own learning is pointed out.

**Keywords:** self-determination theory; controlling behaviours; undiscipline; motivation

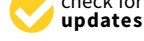

## 1. Introduction

The educational environment is one of the contexts which most influences a student's training and education; therefore, it is important to encourage positive experiences within physical education (PE) classes. The quality of these experiences will prove crucial in inspiring physical activity (PA) in students in the future [1]. It is important for students to feel a sense of connection with their activities, knowing what they like the most and what provides the most enjoyment, providing them an enriched movement experience. This also will have a positive resonance on the development of PE classes, as shown in previous studies [2], where physical activity programs and participation in classes, both within school as well as extracurricular, have been shown to inspire students to feel more motivated towards PE classes.

One of the teaching dilemmas is finding the balance between the use of less controlling and active teaching–learning styles or more controlling and direct instruction methodologies [3]. It is true that group behaviour can influence the type of teaching style used by the teacher and vice versa [4]. However, the goal of every teacher is to get their student to learn and to feel interest in what they are learning [5]. For this, it is essential to have discipline behaviours from the students. Del-Villar [6], defined disciplined behaviours as the ensemble of norms which regulate classroom coexistence, facilitating a sense of order

and organization when performing learning tasks. According to different studies, disciplined behaviours in classrooms [7,8] are crucial in order to achieve a successful process of teaching and learning, as is the motivational predisposition towards PE classes [9–12]. According to Sáenz-López [13], the teaching–learning process of classes encompasses numerous aspects, one of the most important being organization and discipline as, without the same, the objectives would not be attained. For this reason, discipline is considered one of the most important pedagogical aspects and both a challenge and issue of great concern within educational contexts [10,14–16]. Without effective discipline, the actual process of teaching–learning becomes ineffective [17,18].

So, what is the most favourable teaching–learning style to achieve this discipline? A teacher's style of controlling behaviours in teaching–learning is a continuum which progresses from a more controlling style, in which excessive personal control and intimidation predominates (usually at the beginning of contact with the group of students), towards a maximal support of autonomy (when the group is already known), where the student is allowed to make decisions as well as to assume a certain level of responsibility [19,20]. DeMeyer et al. [21] or Reeve & Tseng [22], determined that the use of more controlling styles in classrooms can trigger negative consequences in students, such as undiscipline behaviours. Recent studies such as that of Behzadnia et al. [23] demonstrated a negative relationship between the perception of the student about the teacher's controlling styles and the well-being of the students in the PE classes. In other words, the style used by the teacher could influence not only discipline but also the student's motivation towards PE classes. Moreno-Murcia et al. [24] and Fin et al. [25] affirmed that the controlling behaviours that the teacher uses will have an important impact on student motivation.

In order to analyse the motivational processes in the classroom, the present study is based on the postulates of Self-Determination Theory (SDT) [26,27]. SDT [28–31] is a macro-theory of human motivation which analyses the extent to which human behaviours are volitional or self-determined, in short, the degree to which people perform their actions voluntarily or by their own choice [32,33]. SDT establishes that motivation is a continuum, characterised by different types of self-determination and, therefore, from more to less self-determined. We find intrinsic motivation, integrated regulation, identified regulation, introjected regulation, external regulation, and amotivation. Deci & Ryan [27] have established that this theory is based on the fact that human behaviour is driven by three primary and universal psychological needs: autonomy, which reflects a desire to be committed to activities by one's own free choice, as the origin of one's own behaviour [26,34–36]; competence, to experience a sense of competence by producing desired results and preventing undesired events [26,35,37,38]; and a relationship with others, which refers to the effort to relate and be concerned about others as well as feeling that others have an authentic relationship with oneself and being satisfied socially [29], considering these needs as being innate, universal, and essential for health and wellbeing [30,39].

On the basis of SDT, Vallerand [40,41], and Vallerand and Rousseau [40–42] developed the Hierarchical Motivation Model (HMM). The HMM establishes that the social aspects of the environment influence motivation according to whether a series of Basic Psychological Needs (BPN) are fulfilled (autonomy, competence, and relatedness), the satisfaction of which increases the level of intrinsic motivation [27,39] and will produce positive consequences on an affective, cognitive, and behavioural level [42]. In contrast, the lack of satisfaction of the same progressively increases extrinsic motivation and, ultimately, amotivation, deriving in a series of negative consequences on an affective, cognitive, and behavioural level [41]. Therefore, trying to answer the questions asked above in the present study, the contextual level was analysed, more specifically, based on the PE classes in which our subjects participated, analysing as an antecedent variable the teacher's controlling style and as consequence variables the discipline and undisciplined behaviours of the students.

According to several authors, the perception that students have regarding controlling styles and coercive and authoritarian behaviours facilitates the psychological need to thwart as well as feelings of unease [43–49]. In the same vein, Bartholomew et al. [50] added, in their

longitudinal study, that this perception of the students regarding controlling styles correlated positively with controlled motivation (introjected regulation and external regulation), thwarting unease in subjects [45,51,52]. Thus, a more self-determined motivation increases student satisfaction with classes [53–55], and this trigger in disciplined classes and the importance shown towards PE are greater [56]. Studies such as that of Granero-Gallegos, et al. [57] show that students (especially boys, compared to girls) have a higher rate of undisciplined behaviour, and these in turn are the most unmotivated, in PE classes.

The teaching–learning process carried out by the PE teacher is crucial for students to develop a sense of autonomy and discipline [58,59]. Thus, it will be easier to incorporate physical activity practice into their leisure time. These positive aspects can support the development and consolidation of behaviours related with physical activity [60–62]. Students will feel autonomous if they are given the option to make decisions, without controlling behaviour, on the part of the teacher. This will allow students to satisfy their BPN and to experience more self-determined motivation, which is associated with positive aspects such as improved learning and disciplined behaviours [63,64].

Taking into account the postulates of SDT, the aim of the present study was to examine the relationship between controlling styles of teaching–learning, motivational variables (psychological need thwarting and amotivation), and disciplined and undisciplined behaviours in PE classes among students in secondary school based on HMM [40–42].

Thus, the present hypothesis was that the perception of the pupil regarding the less controlling style (control in the use of rewards) of their teacher would negatively predict the psychological need to thwart (autonomy, competence, and relatedness) and that the perception of the pupils regarding the most controlling teaching–learning style (judging and devaluing) would positively predict the unfulfillment of BPN. This BPN thwarting would positively predict amotivation, while less self-determined motivation would negatively predict disciplinary behaviours in the classroom and positively predict undisciplined behaviours.

## 2. Materials and Methods

This study received approval of the Commission of Bioethics and Biosecurity of the University of Extremadura and followed the guidelines of the Helsinki Declaration. All participants were treated in agreement with the ethical guidelines of the American Psychological Association with regards to participant assent, parent/guardian consent, confidentiality, and anonymity. Moreover, informed written consent was obtained from both the participants and their parents/guardians.

### 2.1. Sample

The study sample comprised 922 students of both sexes (430 male students and 492 female students) in their 3rd and 4th years of secondary school. Several public and charter schools located in the same city of Spain were selected. Nine schools participated in this study with a total of 50 classes. Each class comprised 18–20 students. The ages of the sample ranged between 14 and 18 years (M = 14.95; SD = 0.98). Twelve students were excluded from the study. The exclusion criteria were not answering the majority of questions and unusual response patterns. The type of sampling that was carried out was intentional [65].

### 2.2. Variables and Measurement Tools

To gather evidence of reliability, two indicators of internal consistency were used: Cronbach's Alpha ($\alpha$) and McDonald's Omega ($\omega$). The latter accounts for the lack of tau equivalence (unequal factor loadings). The range of both indexes is between 0 and one, with the highest values providing the most reliable measurements [66]. To gather evidence of structural validity, a Confirmatory Factorial Analysis (CFA) was performed. More information about the estimation method and fit can be found in the Data Analyses section.

2.2.1. Perception of the Teacher's Controlling Behaviours

To examine the perception of the teacher's controlling behaviours, on behalf of the students, we used the Controlling Coach Behaviours Scale (CCBS). This scale was validated by Bartholomew et al. [43] and adapted into Spanish by Castillo et al. [67]. For the purpose of this study, it was also adapted to the educational setting (based on Bartholomew et al. [43]). Specifically, the expression "My coach" was replaced by "My teacher of PE" in the corresponding statements. The CCBS comprises 20 items divided into five factors, each of which comprises four items. Two factors were used in this research: the least controlling, control of the use of rewards (e.g., I am motivated by the praise I receive if I do it well) (results showed acceptable fit for reliability: $\alpha = 0.72$, $\omega = 0.78$) and the most controlling, judging and devaluing (e.g., The teacher undervalues my contributions in class) (results showed acceptable fit for reliability: $\alpha = 0.76$, $\omega = 0.84$). In the CFA, the results showed acceptable fit indexes: $\chi^2 = 584.43$, *df* (degrees of freedom) = 142 ($p < 0.001$), CFI (Comparative Fit Index) = 0.98, TLI (Tucker-Lewis Index) = 0.98, and RMSEA (Root Mean Square Error of Approximation) = 0.06 (90% CI (Confidence interval) = 0.05, 0.06).

2.2.2. Psychological Need Thwarting

To assess the level of psychological need thwarting, the Psychological Need Thwarting Scale (PNTS) was used. This original scale by Bartholomew et al. [68] was validated into Spanish by Sicilia et al. [69], and it was also adapted to the educational setting (based on Cuevas et al. [70]). Specifically, the expression "In my physical exercise" was replaced by "In my PE classes" in the corresponding statements. The questionnaire comprised 12 items divided into three factors: autonomy (e.g., I feel obliged to follow decisions of others in classes) (results showed acceptable fit for reliability: $\alpha = 0.73$, $\omega = 0.80$), competence (e.g., At times, they have said things that make me feel that I am no good at exercising) (results showed acceptable fit for reliability: $\alpha = 0.82$, $\omega = 0.88$), and relatedness (e.g., I feel like other people don't like me) (results showed acceptable fit for reliability: $\alpha = 0.87$, $\omega = 0.93$). Regarding the CFA, the results showed acceptable fit indexes for BPN of autonomy: $\chi^2 = 87.81$, *df* = 32 ($p < 0.001$), CFI = 0.99, TLI = 0.99, and RMSEA = 0.04 (90% CI = 0.03, 0.05).

2.2.3. Amotivation

To measure the levels of self-determined motivation, the Perceived Locus of Causality (PLOC) for PE was used. The original scale was by Goudas et al. [71] and has been validated into Spanish by Moreno et al. [72]. This scale is formed of 20 items, divided into five factors. Each factor comprises four items. In our research, the amotivation factor (e.g., But I don't really know why) was used. Results showed acceptable fit for reliability: $\alpha = 0.76$, $\omega = 0.85$. Concerning the CFA, our results revealed acceptable fix indexes: $\chi^2 = 269.79$, *df* = 62 ($p < 0.001$), CFI = 0.96, TLI = 0.97, and RMSEA = 0.06 (90% CI = 0.05, 0.07).

2.2.4. Discipline–Undisciplined Behaviours of Students

To measure the presence of disciplined or undisciplined behaviours in the classroom, the Inventory of Disciplined–Undisciplined Behaviour in Physical Education (ICDIEF) was used. This questionnaire was developed and validated by Cervelló et al. [73], comprising 20 items divided into two factors. Each factor comprises 10 items: discipline (e.g., You try your best so that there is a good atmosphere in class) (results showed acceptable fit for reliability: $\alpha = 0.80$, $\omega = 0.89$) and undiscipline (e.g., You tend to use the material inappropriately). Results showed acceptable fit for reliability: $\alpha = 0.84$, $\omega = 0.92$. Regarding the FAC, the results revealed acceptable fit indexes: $\chi^2 = 293.20$, *df* = 76 ($p < 0.001$), CFI = 0.98, TLI = 0.97, and RMSEA = 0.06 (90% CI = 0.05, 0.06).

In all questionnaires used, all the items are responded to using a Likert scale of five points, ranging from value 0, totally in disagreement to 5, totally agree.

*2.3. Procedure*

Once the study aims were defined, the appropriate measurement tools in order to gather the study data was selected. We subsequently prepared a dossier with the same and also gathered additional pertinent data, such as the age, academic year, and the high school attended. Thereafter, we contacted the different high schools explaining the aim of our study and asking whether they were interested in participating. Upon acceptance, they were given an informed consent form for the parents to complete, as the students concerned were under the age of 18 years. We allowed a period of time for them to supply the necessary authorization and, thereafter, arranged when to visit the specified high schools and to deliver the questionnaires. We ensured that the PE teacher was not present at the time. The questionnaires were applied between March and June 2017. They were completed within school hours, and completion of the questionnaire was individual. The amount of time necessary for completing the questionnaires was approximately 30 min.

*2.4. Statistical Analyses and Treatment of Data*

First, we computed mean and standard deviation, correlation, and intraclass correlation (an indicator of not independence). The normality tests were realized with the Kolmogorov–Smirnov test. It was determined that most of the variables were normal, due to which parametric statistical tests were applied. The study hypotheses were analysed via Structural Equation Modelling (SEM). It is important to note that data may not be independent, since students were nested within classes and schools [74]. This may lead to worse fit and an underestimation of standard errors. To statistically correct this nesting, we used a sandwich-type estimator [75].

To test the indirect effects, we used the delta method [76]. The hypothesised indirect effects were the effect of the assignment of responsibility in the evaluation on satisfaction with the physical education classes, through the BPN and autonomous motivation.

We used the following goodness of fit indexes: chi-square ($\chi^2$), *df*, significance (*p*), RMSEA, CFI, and the TLI.

Regarding the estimation method, considering that the answers of the participants were obtained using a Likert scale and their answers were ordered categorically, we decided to use weighted least square mean and variance adjusted (WLSMV) as the estimation method. The WLSMV is more accurate than the maximum likelihood [77], as it does not require multivariable or univariate normality [78]. For descriptive analyses, the statistical program SPSS 21.0 was used, and for the SEM, calculations were done with Mplus 8.3.

## 3. Results

*3.1. Reliability Analysis and Descriptive Statistics*

Table 1 presents the descriptive statistics, correlations, and intraclass correlation of study variables. The table displays the mean (M) and the standard deviation (SD) of all the study variables observing that, regarding the controlling behaviour of the PE instructor, the highest value of the mean was for control in the use of rewards. Regarding the Psychological Need Thwarting Scale for thwarting of autonomy need within the PLOC for autonomous motivation and regarding the Inventory of Behaviours of discipline–undiscipline in PE, the highest value was for discipline behaviours in PE classes. With regard to ICC, the results showed low values.

**Table 1.** Means, standard deviations, and correlations among variables.

| Variable | M | SD | ICC | 1 | 2 | 3 | 4 | 5 | 6 |
|---|---|---|---|---|---|---|---|---|---|
| CCBS | | | | | | | | | |
| 1. Control of the use of rewards | 2.26 | 0.95 | 0.02 | - | | | | | |
| 2. Judging and Devaluing | 1.89 | 0.97 | 0.04 | 0.60 | - | | | | |
| PNTS | | | | | | | | | |
| 3. Psychological need thwarting of Autonomy | 2.19 | 1.02 | 0.01 | 0.35 | 0.45 | - | | | |
| PLOC | | | | | | | | | |
| 4. Amotivation | 1.88 | 0.92 | 0.03 | 0.14 | 0.28 | 0.30 | - | | |
| ICDIEF | | | | | | | | | |
| 5. Discipline | 4.06 | 0.66 | 0.03 | −0.08 | −0.21 | −0.25 | −0.33 | - | |
| 6. Undiscipline | 1.92 | 0.79 | 0.03 | 0.35 | 0.47 | 0.38 | 0.31 | −0.46 | - |

Note: M, Mean; SD, Standard Deviation; and ICC; Intraclass Correlation. All correlations were significant with $p < 0.001$.

### 3.2. Structural Equation Modelling

The model suggested that the students' perception of the teacher's less controlling style (control in the use of rewards) would negatively predict the psychological need to thwart while the perception of students about the teacher who had a more controlling style (judging and devaluing) would positively predict the psychological need thwarting. This psychological need to thwart would positively predict amotivation, while this would negatively predict disciplined behaviours in the classroom and positively predict those undisciplined behaviours not being generated. we encountered problems of convergence and were unable to derive results for the latter.

Sometimes basic needs correlate so highly [79] that applied researchers face multicollinearity, which is a main cause of convergence issues [80]. To solve this problem, researchers have focused on the main SDT need, autonomy, or inclusion of all needs in one factor. In this research, we preferred to focus just on autonomy, more specifically, the thwarting of autonomy need, as it is the construct that distinguishes SDT research from other theories (i.e., self-efficacy, attachment, etc.).

The results of the new proposed model can be seen in Figure 1. This shows six latent variables, with a total of 27 observed variables and each latent variable having a minimum of three items.

In this model, the intention was to determine the predictive variables of the disciplined and undisciplined behaviours in PE classes, based on the perception of the controlling styles of the teacher and the motivational variables (thwarting of autonomy need and amotivation). The results are displayed in Figure 1.

The contribution of each of the factors to the prediction of other variables was examined via standardised regression weights. Therefore, the thwarting of autonomy need was negatively and significantly predicted by the perception of a less controlling style on behalf of the teacher, control in the use of rewards (β = −0.20) (−0.39, −0.01), and positively and significantly with the more controlling style of the teacher, judging and devaluing (β = 0.98) (0.79, 0.16). In turn, the thwarting of autonomy need positively and significantly predicted amotivation (β = 0.78) (0.71, 0.85), and this negatively and significantly predicted disciplined behaviour in the classroom (β = −0.50) (−0.57, −0.43), and positively and significantly predicted undisciplined behaviour during classes (β = 0.79) (0.73, 0.85). Besides, a positive and significant relationship was observed among the teachers' different controlling styles (β = 0.83) (0.79, 0.85).

In addition, the results of the model presented acceptable fit indexes: $\chi^2$ = 797.76, *df* = 317, *p* = 0.00, CFI = 0.95, TLI = 0.94, and RMSEA = 0.04 (IC 90% = 0.04, 0.04). Below, in Table 2, the results of the indirect standardised effects of the structural equation modelling are displayed.

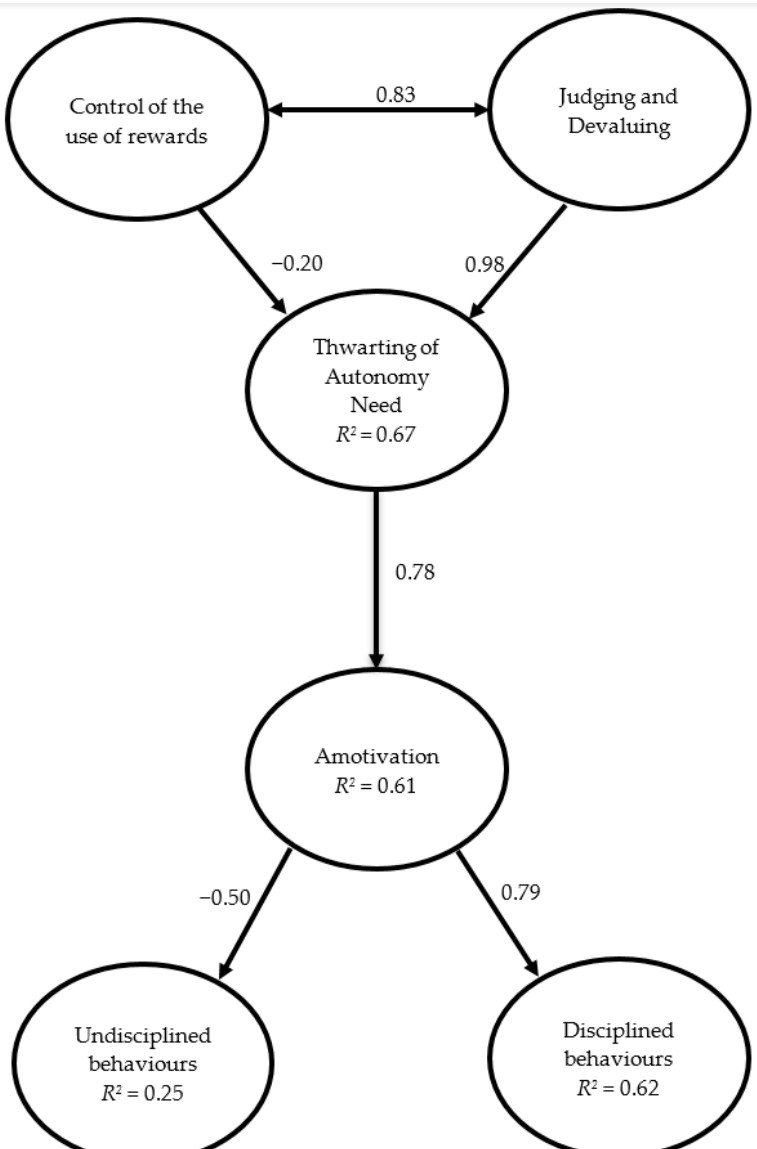

**Figure 1.** Structural Equation Modelling (SEM): Prediction of disciplined and undisciplined behaviours based on the perception of students regarding the controlling styles of the teacher and motivational variables. All parameters are standardized and are statistically significant ($p < 0.05$).

**Table 2.** Indirect effects in structural equation modelling.

| Variables | | | Effects | *p* | CI |
|---|---|---|---|---|---|
| Control of the use of rewards | → | Amotivation | −0.16 | 0.04 | −0.31, −0.00 |
| Judging and Devaluing | → | Amotivation | 0.77 | 0.00 | 0.61, 0.92 |
| Control of the use of rewards | → | Disciplined behaviours | 0.08 | 0.04 | 0.00, 0.16 |
| Judging and Devaluing | → | Disciplined behaviours | −0.39 | 0.00 | −0.46, −0.31 |
| Control of the use of rewards | → | Undisciplined behaviours | −0.12 | 0.04 | −0.24, −0.01 |
| Judging and Devaluing | → | Undisciplined behaviours | 0.60 | 0.00 | 0.49, 0.70 |
| Thwarting of Autonomy Need | → | Disciplined behaviours | −0.40 | 0.00 | −0.44, −0.35 |
| Thwarting of Autonomy Need | → | Undisciplined behaviours | 0.62 | 0.00 | 0.57, 0.66 |

Note: CI, Confidence interval; *p*, significance index.

As can be observed in Table 2, the indirect standardised effects revealed that control in the use of rewards had a negative, indirect effect on amotivation and undisciplined behaviours. Regarding the variable judging and devaluing, we observed indirect positive effects on amotivation and undisciplined behaviours and negative effects on disciplined behaviours. Concerning the thwarting of autonomy need, negative indirect effects were found towards behaviours of discipline and positive effects on behaviours of undiscipline.

## 4. Discussion

It is interesting to predict what type of teaching–learning and motivational variables are determinant to promote discipline behaviours in the classroom. The initial hypothesis was the following: "The student's perception of the less controlling style (control of the use of rewards) of the teacher will negatively predict the psychological need thwarting (autonomy, competence, and relatedness), and the students' perception of a teacher's more controlling style (judging and devaluing) will positively predict the psychological need thwarting. This, will positively predict amotivation, and this will negatively predict disciplinary behaviours in the classroom and positively predict undiscipline behaviours". This is only partially fulfilled in our study, as the only latent variable was the thwarting of autonomy need, of the three hypothesised psychological needs to thwart. This model obtained acceptable fit indexes, observing that perception of the less controlling style (control in the use of rewards) negatively and significantly predicted the thwarting of autonomy need. In addition, the perception of a more controlling style (judging and devaluing) predicted the same positively and significantly. The thwarting of autonomy need positively and significantly predicted amotivation. This negatively and significantly predicted disciplined behaviours and positively and significantly predicted undisciplined behaviours in the classroom.

Similar data to those obtained in the present study are found in other research studies within the educational context. We consider that the strength and novelty of this prediction is the determination of the control of the discipline of the students through less controlling teaching–learning styles.

Blanchard et al. [81] and Behzadnia et al. [23] found that a more controlling style predicted the thwarting of autonomy need. Similar results were reported by Balaguer et al. [82], who found that more controlling teaching–learning styles favoured the psychological need to thwart. Likewise, Haerens et al. [83] found that more controlling styles were related to the frustration need of autonomy, deriving in a less self-determined type of motivation in the classroom, even leading to amotivation. Also, a study by Abós et al. [84], among a sample of students in their first and second years of secondary school, revealed that it is necessary to avoid the use of more controlling styles in order to prevent the psychological need to thwart and, thus, the appearance more challenging behaviours on behalf of the students. More recent studies, such as that performed by Bartholomew, et al. [50], Fin et al. [25], and Moreno-Murcia et al. [24], found that more controlling teaching–learning styles are related to less satisfaction of BPN as well as less self-determined type of motivation. Strategies such as helping students to be more responsible in their behaviour will present more self-determined forms of motivation compared to strategies based on the use of punishments or rewards [85]. Therefore, the results found in the present study support the HMM, suggesting that reducing the thwarting of autonomy need reduces less self-determined forms of motivation, causing disciplinary behaviours among the students.

We asked ourselves what was the most appropriate teaching–learning style to achieve disciplined behaviours, and it was noted how the student´s self-determined motivation explains these behaviours. Similar studies performed by DeMeyer et al. [21] and by Reeve and Tseng [22] showed that the use of more controlling styles can lead to negative behaviours in students. Likewise, analogous to our results also, Haerens et al. [83] found that the use of more controlling styles, positively and significantly predicted the frustration need of autonomy which also positively and significantly predicted amotivation and this positively and significantly predicted the appearance of undisciplined behaviours

in classrooms. In summary, several works have found a relationship between the use of more or less controlling teaching–learning styles with regards to the satisfaction or thwarting of autonomy need [15,84,86] as well as positive consequences [87–90] or negative consequences for processes of teaching–learning [21,22,57,83] within PE classes. In line with the theoretical framework [41], it can be observed that there are certain backgrounds, such as the controlling style of the teacher, which can predict the thwarting of autonomy need. This, in turn, can make motivation less self-determined, leading to amotivation, which can trigger negative consequences for the teaching–learning processes and, more specifically, the appearance of undisciplined behaviours in PE classes.

Moreno et al. [91] found that the environment where the task is performed was a positive predictor of intrinsic motivation in order to maintain discipline in the classrooms. However, the structuring of PE classes is a concern for teachers when applying less controlling teaching–learning styles [64]. Therefore, it is very important to equip teachers with strategies to make the application of these styles effective, and studies such as Cheón et al. [64] are a good example of how to carry out and intervention with teachers and students to improve the application and structuring of more autonomous teaching–learning styles. Also, for application of these styles, Liu et al. [48] determined the importance for the teacher that the student is intrinsically motivated, as this will help them to propose and apply these strategies. It is important that teachers, in addition to using strategies to improve student autonomy, consider fun activities adapted to all levels [92].

In addition to the importance of supporting autonomy by the teacher, it is necessary to take into account the controlling behaviours by the teacher and to try to minimise them. Tiga et al. [93] highlighted the importance of minimising the controlling behaviours instead of focusing only on fostering support for autonomy, which in turn was related to positive consequences in students, after demonstrating that students are sensitive to controlling behaviours and even experiencing support for autonomy from their teachers. Along the same lines, Koka et al. [94] demonstrated, through a model of structural equations, that the perception of controlling behaviours by the students negatively affected the amount of physical activity carried out in leisure time. More recently [95], they demonstrated a negative relationship between controlling behaviours by PE teacher with the intention and participation of the student to carry out physical activity outside of school.

One of the aims of teachers should be the use of less controlling styles, based on avoiding the thwarting of autonomy need. This should be achieved using different motivational strategies, including granting students more responsibilities, allowing them to participate more in their learning process, knowing what they do and why, valuing their progress, proposing different tasks and encouraging students to select the one that aligns most with their interests, and allowing them to make decisions regarding the performance of activities. This would avoid the psychological need to thwart and, thus, increase more self-determined forms of motivation, thus facilitating the appearance of disciplined behaviours in the classroom.

With the results of this study, it can be determined that the more controlling styles favours the psychological need to thwart, less self-determined motivation, and undisciplined behaviours, thus responding to one of the main problems that currently exist in classrooms. Therefore, we highlight the importance of both the application of strategies by teachers that promote autonomy and decision-making by students. However, we must take into account a series of limitations of this study in relation to the social environment in which students develop and how this influences their motivation as well as their classroom behaviours. It would not be correct to point to the application of controlling or non-controlling teaching–learning styles as the sole cause of discipline or undiscipline in PE classes, since there are many other determining factors. The feeling of belonging to a certain environment and/or social connection between students and teachers is a fundamental aspect for motivation [96]. In addition, the cultural norms of the home are also important factors: parents promoting PA is beneficial for motivation of students [96]. This

motivation may also be conditioned by aspects such as being afraid to perform a physical test [83], troubles relating to others, or lack of fun [97].

If these variables above are taken into account in the teaching–learning process, it will undoubtedly help us understand many of the undisciplined behaviours of the students, so future lines of research should consider the inclusion of these variables. In addition to the psychosocial variables discussed above, other limitations of the present study were the lack of comparisons according to the age and gender of students. In addition, the study was based on questionnaires alone, rather than being performed in the classroom or as an observational or longitudinal study. Therefore, we recommend the performance of future analyses comparing these variables according to the students' age and gender as well as using systematic observation questionnaires during classes, accompanied by interviews. We also would like to highlight that testing of the studied relationships in our study could add causal evidence if tested in a longitudinal analysis. Furthermore, intervention programs are needed, focused on satisfaction of the BPN for autonomy to encourage the appearance of disciplined behaviours among students.

## 5. Conclusions

In conclusion, in classrooms, the use of controlling styles of teaching–learning on behalf of teachers should be avoided via the promotion of the BPN of autonomy and by assigning greater responsibilities to students. This will result in less amotivation and undisciplined behaviours among students and in improvements of teaching–learning processes in the classroom.

**Author Contributions:** Conceptualization, M.L.-R. and R.J.-C.; methodology, M.L.-R., M.B., and R.J.-C.; software, R.J.-C and J.J.L.G.-V.; validation, M.L.-R. and R.J.-C.; formal analysis, M.L.-R., R.J.-C., and J.J.L.G.-V.; investigation, M.L.-R., M.B., and R.J.-C.; resources, M.L.-R. and R.J.-C.; data curation, M.L.-R., J.J.L.G.-V., and R.J.-C.; writing—original draft preparation, M.L.-R.; writing—review and editing, M.L.-R.; visualization, M.L.-R.; supervision, R.J.-C.; project administration, R.J.-C. All authors have read and agreed to the published version of the manuscript.

**Funding:** This study was carried out thanks to the contribution of the Ministry of Economy and Infrastructure of the Council of Extremadura, through the European Regional Development Fund—A way to make Europe. (GR18129). We would like to specify that the financial assistance is only for Ruth Jiménez Castuera.

**Institutional Review Board Statement:** The study was conducted according to the guidelines of the Declaration of Helsinki, and approved by Commission of Bioethics and Biosecurity of the University of Extremadura (10/12/2015).

**Informed Consent Statement:** Informed consent was obtained from all subjects involved in the study.

**Data Availability Statement:** The data presented in this study are available on request from the corresponding author. The data are not publicly available due to ethical reasons.

**Acknowledgments:** We wish to thank all the students for their selfless participation in the present study.

**Conflicts of Interest:** The authors declare no conflict of interest. The funders had no role in the design of the study; in the collection, analyses, or interpretation of data; in the writing of the manuscript; or in the decision to publish the results.

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
