# Peer review of "Predictive Model for Amotivation and Discipline in Physical Education Students Based on Teaching–Learning Styles"

_sustainability, doi:10.3390/su13010187_

Round 1
Reviewer 1 Report
There is a mistake in line 44 "to fell interest", you maybe wanted to say "to feel interest".
Each word of the abbreviations of the jourals´names must have a dot at the end of each one.
Most references must be updated, as more than 50% of them are more than 5 years old. Generally speaking, 70% of references must be current, i.e. from 5 years ago to the present day. I advise to review this aspect in depth
Author Response
First of all, we would like to thank all the comments and suggestions of both the editor and the reviewer, which helps us to improve and give a higher quality to the work done and presented. Then each comment is answered point by point. We hope to answer cleary to the comments and suggestions of the editor and the reviewer. If it were necessary to clarify any concept or point, let us know.
There is a mistake in line 44 "to fell interest", you maybe wanted to say "to feel interest".
Response: Thank you very much for the correction. We have modified it in the text.
Each word of the abbreviations of the journal´s names must have a dot at the end of each one.
Response: We have put the points after each abbreviation.
Reviewer 2 Report
Introduction. The part of the theoretical framework presents all the variables to be studied in a structured and solid way, being the current and corresponding references according to the subject. One of the variables to study is motivation, which in my point of view is little treated in this one. It is recommended to expand this variable a little more since the above is short and not complete to inform the reader about it. Material and method It is recommended to put the ethics committee followed for this research as it needs to be mentioned. Not only with the Helsinki declaration. Please indicate it. How was the sample selected? Was it for convenience? Why those centers and not others? What selection criteria did they use to do it? Is it a normal or non-normal sample? What times were used to pass the questionnaires? Was it collectively passed to all groups at the same time? Was it before or after free time? Could they talk to each other while doing it? The data treatment section presents in form and structure all the elements necessary for a good analysis, leaving no doubt regarding the use of the different statistical studies presented by the research. Regarding the results, everything is considered correct in format and form. It is recommended to use all digits (0.08) and not (.08) in the document. Discussion It argues the results with different points of view that makes the final document or the report complete, does not present gaps and the connections are correct. It would be recommended not to use personal forms, but rather the impersonal one. Line 352 “We asked”. Regarding the references, it can be seen that a large part of them are current but there are some somewhat outdated that it is recommended to change and update.
Author Response
First of all, we would like to thank all the comments and suggestions of both the editor and the reviewer, which helps us to improve and give a higher quality to the work done and presented. Then each comment is answered point by point. We hope to answer cleary to the comments and suggestions of the editor and the reviewer. If it were necessary to clarify any concept or point, let us know.
Introduction. The part of the theoretical framework presents all the variables to be studied in a structured and solid way, being the current and corresponding references according to the subject. One of the variables to study is motivation, which in my point of view is little treated in this one. It is recommended to expand this variable a little more since the above is short and not complete to inform the reader about it.
Material and method It is recommended to put the ethics committee followed for this research as it needs to be mentioned. Not only with the Helsinki declaration. Please indicate it.
Response: We have added Commission of Bioethics and Biosecurity of the University of Extremadura (Line 126).
How was the sample selected? Was it for convenience?
Response: The type of sampling that was carried out was intentional [65]. (Line 137)
Why those centers and not others? What selection criteria did they use to do it?
Response: We got in touch with all the city centers. Those who wanted to participate in the study were selected.
Is it a normal or non-normal sample?
Response: We have added “The normality tests were realized with the Kolmogorov-Smirnoff test. It was determined that most of the variables were normal, due to which parametric statistical tests were applied.”(Line 202)
What times were used to pass the questionnaires? Was it collectively passed to all groups at the same time? Was it before or after free time? Could they talk to each other while doing it?
Response: We have added “We ensured that the PE teacher was not present at the time. The questionnaires were applied between March and June 2017. They were completed within school hours and the completion of the questionnaire was individual. The amount of time necessary for completing the questionnaires was approximately 30 minutes.” (Line 196)
The data treatment section presents in form and structure all the elements necessary for a good analysis, leaving no doubt regarding the use of the different statistical studies presented by the research. Regarding the results, everything is considered correct in format and form. It is recommended to use all digits (0.08) and not (.08) in the document.
Response: We have added all the digits throughout the document.
Discussion It argues the results with different points of view that makes the final document or the report complete, does not present gaps and the connections are correct. It would be recommended not to use personal forms, but rather the impersonal one. Line 352 “We asked”.
Response: We have corrected the writing style throughout the manuscript.
Regarding the references, it can be seen that a large part of them are current but there are some somewhat outdated that it is recommended to change and update.
Response: We have updated some references, and introduced new ones such as references 93, 94 and 95
Reviewer 3 Report
Comments to the Authors
I would like to thank the Authors for conducting this research as it presents a very important and interesting topic in the context of physical education. I have read this paper with much interest. Please see some minor concerns related to this paper.
Title
The title contains most of the key features of the article. Also, I think it should be specified that it is “amotivation” in the title?
Abstract
The abstract is very well written and all important information is provided for the reader.
Line 17. I think it should be specified that positive effect on which variable was meant here? I believe Authors wanted to make a ‘general statement’, but it should be precise that “positive effect on …”. Because controlling behaviours might have a positive effect on need thwarting, but it is not that ‘positive thing to have’ at all.
Line 18. I suggest that Authors should use “controlling behaviours” because “interpersonal behaviours” is usually used when both autonomy support and controlling behaviours are measured. It is clearer for the reader if you use only controlling behaviours as this is what you measured. Please also specify that it is students’ perceptions of their physical education teachers’ controlling behaviour, not the actual (or objectively measured) physical education teachers’ behaviour. Please specify that it is amotivation, not just motivation.
Line 21. I am wondering what is less controlling discipline styles? Students perceptions of their physical educations teachers controlling behaviours were measured by using CCBS that comprised two subscales in the current study. I later see that you theorize that “Control of the use of rewards” as less controlling. I suggest to use only the name of this subscale so that it would be clear for everyone that what is meant. I also suggest that use throughout term “controlling behaviours” and do not mix it with “interpersonal behaviours”, “styles” or “discipline styles”.
Line 23. Please specify what is “this” that predicted positively amotivation.
Line 24. Please specify what teachers should avoid. One subscale of CCBS predicted positively autonomy need thwarting and another predicted negatively autonomy need thwarting. However, the conclusion is that physical education teachers still should avoid both behaviours?
Keywords: please note that it is suggested to not use the same terms as keywords that have been already used in the title of the paper.
Introduction
The authors provide adequate review of the existing literature. Specifically, the authors provide logical progression from existing knowledge that leads their research question and highlight the important patterns. Also, the authors provide sufficient understanding what the paper is about. Most of the recent and relevant studies are included. Overall, the introduction is clear and concise.
Specific comments:
Line 33. Why is physical education with capital letters?
Page 2, line 80. I believe it should be need for relatedness?
Line 89. “Or mediators” does not seem to fit here.
Line 109. “Students” needs a capital letter.
Line 121. Please note that the unfulfillment is not the same as thwarting. Also, I suggest not to use term dissatisfaction. Please use the term what was measured, psychological need thwarting.
Materials and Methods
Specific comments:
For the results of CFA of each scale, please also report the value of degrees of freedom (df).
Line 145. Please use throughout the manuscript term “controlling behaviours”.
Line 158. I suggest to not use “levels of”. It needs specific additional data analysis to find out the “levels of”.
Line 170. For clarity, this title 2.2.3. should read “Amotivation” as that was what you actually measured.
I am also left wondering what was the statistical program used in the current study? That is usually reported in the section of statistical analyses.
Line 209. I believe that degrees of freedom should be df, not gl?
Line 234. I would keep using amotivation and I suggest not to use “less self-determined form of motivation” as a synonym because it might be confusing for the reader.
Line 240. Is there any rationale why it was chosen to use only need for autonomy and not a general factor that contains all needs?
Figure 1. I would like to see more specific p-values of the coefficients and not just p < .05. I believe some of the values are even p < .001? This should be specified.
Line 289 reads that all parameters are standardised. However, on line 298 I read CI value that is 1.16.
Line 304. I am not sure if gl=df? Also, can a p value be exactly =.00?
Table 2. Not sure if it is valid to write “-.00”.
Discussion
The authors have discussed the results from multiple angles and placed it into proper context without being overinterpreted. The authors link their findings to previous studies.
Specific comments:
Line 318. “ starts here but I cannot find where it ends.
Line 336. I suggest that you should use “and” instead of “or”.
Line 339. I believe “Hearens” is a typo. Also, please note that Haerens et al. (2015) examined need frustration, not need thwarting.
Lines 336-351: I believe a study by Tilga et al. (2019) could provide a strong rationale here to highlight the importance of minimizing controlling behaviour, rather than an exclusive focus on enhancing autonomy-supportive behaviour, to enhance students’ adaptive outcomes. The reason is that students are sensitive to controlling behaviours even if they experience autonomy-supportive behaviour from their PE teachers.
Tilga, H., Hein, V., Koka, A., Hamilton, K., & Hagger, M. S. (2019). The role of teachers’ controlling behaviour in physical education on adolescents’ health-related quality of life: Test of a conditional process model*. Educational Psychology, 39(7), 862–880. https://doi.org/10.1080/01443410.2018.1546830
Lines 359-361: I believe a recent study by Koka et al. (2019) is very relevant here (see Figure 1 in a study by Koka and colleagues).
Koka, A., Tilga, H., Kalajas-Tilga, H., Hein, V., & Raudsepp, L. (2019). Perceived Controlling Behaviors of Physical Education Teachers and Objectively Measured Leisure-Time Physical Activity in Adolescents. International Journal of Environmental Research and Public Health, 16(15), 2709. https://doi.org/10.3390/ijerph16152709
Author Response
First of all, we would like to thank all the comments and suggestions of both the editor and the reviewer, which helps us to improve and give a higher quality to the work done and presented. Then each comment is answered point by point. We hope to answer cleary to the comments and suggestions of the editor and the reviewer. If it were necessary to clarify any concept or point, let us know.
I would like to thank the Authors for conducting this research as it presents a very important and interesting topic in the context of physical education. I have read this paper with much interest. Please see some minor concerns related to this paper.
Title
The title contains most of the key features of the article. Also, I think it should be specified that it is “amotivation” in the title?
Response: We have specified amotivation in the title.
Abstract
The abstract is very well written and all important information is provided for the reader.
Line 17. I think it should be specified that positive effect on which variable was meant here? I believe Authors wanted to make a ‘general statement’, but it should be precise that “positive effect on …”. Because controlling behaviours might have a positive effect on need thwarting, but it is not that ‘positive thing to have’ at all.
Response: We have added “positive effect on students motivation and discipline” (Line19).
Line 18. I suggest that Authors should use “controlling behaviours” because “interpersonal behaviours” is usually used when both autonomy support and controlling behaviours are measured. It is clearer for the reader if you use only controlling behaviours as this is what you measured. Please also specify that it is students’ perceptions of their physical education teachers’ controlling behaviour, not the actual (or objectively measured) physical education teachers’ behaviour. Please specify that it is amotivation, not just motivation.
Response: We have added “student´s perceptions of the controlling behaviours of their physical education teacher, together with amotivation and discipline styles from the Self-Determination Theory.” (Line 20)
Line 21. I am wondering what is less controlling discipline styles? Students perceptions of their physical educations teachers controlling behaviours were measured by using CCBS that comprised two subscales in the current study. I later see that you theorize that “Control of the use of rewards” as less controlling.
Response: Yes, we do mean “Control of the use of rewards”, we have added it to clarify it on line 23.
I suggest to use only the name of this subscale so that it would be clear for everyone that what is meant. I also suggest that use throughout term “controlling behaviours” and do not mix it with “interpersonal behaviours”, “styles” or “discipline styles”.
Response: We have unified the term controlling behaviors, throughout the text.
Line 23. Please specify what is “this” that predicted positively amotivation.
Response: the thwarting of autonomy need. (Line 25).
Line 24. Please specify what teachers should avoid. One subscale of CCBS predicted positively autonomy need thwarting and another predicted negatively autonomy need thwarting. However, the conclusion is that physical education teachers still should avoid both behaviours?
Response: We have specified that teachers should avoid controlling behaviors such as judging and devaluing, both on line 24 and on line 28.
Keywords: please note that it is suggested to not use the same terms as keywords that have been already used in the title of the paper.
Response: We changed the keywords.
Introduction
The authors provide adequate review of the existing literature. Specifically, the authors provide logical progression from existing knowledge that leads their research question and highlight the important patterns. Also, the authors provide sufficient understanding what the paper is about. Most of the recent and relevant studies are included. Overall, the introduction is clear and concise.
Specific comments:
Line 33. Why is physical education with capital letters?
Response: We have put it in lowercase.
Page 2, line 80. I believe it should be need for relatedness?
Response: We have removed the phrase “which implies a desire for the individual to effectively interact with the environment”, since as you point out, it is related to relatedness
Line 89. “Or mediators” does not seem to fit here.
Response: We have removed it.
Line 109. “Students” needs a capital letter.
Response: We have modified it (Line 109).
Line 121. Please note that the unfulfillment is not the same as thwarting. Also, I suggest not to use term dissatisfaction. Please use the term what was measured, psychological need thwarting.
Response: We have modified it. (Line 121).
Materials and Methods
Specific comments:
For the results of CFA of each scale, please also report the value of degrees of freedom (df).
Response: We have entered this data within each instrument. In the Material and Methods section.
Line 145. Please use throughout the manuscript term “controlling behaviours”.
Response: We have modified it throughout the manuscript.
Line 158. I suggest to not use “levels of”. It needs specific additional data analysis to find out the “levels of”.
Response: We have removed “levels of”.
Line 170. For clarity, this title 2.2.3. should read “Amotivation” as that was what you actually measured.
Response: We have modified the subtitle as suggested by the reviewer.
I am also left wondering what was the statistical program used in the current study? That is usually reported in the section of statistical analyses.
Response: It was a mistake. We have entered this information on line 218. “For descriptive analyses, the statistical program SPSS 21.0 was used, and for the SEM, calculations were done with Mplus 8.3”.
Line 209. I believe that degrees of freedom should be df, not gl?
Response: It was a translation error, we have modified it throughout the manuscript.
Line 234. I would keep using amotivation and I suggest not to use “less self-determined form of motivation” as a synonym because it might be confusing for the reader.
Response: We have modified it (Line 237).
Line 240. Is there any rationale why it was chosen to use only need for autonomy and not a general factor that contains all needs?
Response: Because we have based on the original scale: Bartholomew, K.J.; Ntoumanis, N.; Ryan, R.M.; Bosch, J.A.; Thogersen-Ntoumani, C. Self-determination theory and diminished functioning: The role of interpersonal control and psychological need thwarting. Pers. Soc. Psychol. Bull. 2011, 37, 1459-1473. http://dx.doi.org/10.1177/0146167211413125.
Figure 1. I would like to see more specific p-values of the coefficients and not just p < .05. I believe some of the values are even p < .001? This should be specified.
Response: They are all less than 0.05. There is none less than 0.01, so that information has been put.
Line 289 reads that all parameters are standardised. However, on line 298 I read CI value that is 1.16.
Response: We have checked it and it was a transcription error. The value is 0.16. We have modified it.
Line 304. I am not sure if gl=df? Also, can a p value be exactly =.00?
Response: We have modified gl for df throughout the manuscript. The exact value was 0.0001, so we have used 0.00.
Table 2. Not sure if it is valid to write “-.00”.
Response: We have put -0.00.
Discussion
The authors have discussed the results from multiple angles and placed it into proper context without being overinterpreted. The authors link their findings to previous studies.
Specific comments:
Line 318. “ starts here but I cannot find where it ends.
Response: We have put the quotes where it ends “the student’s perception of the less controlling style (control of the use of rewards) of the teacher, will negatively predict the psychological need thwarting (autonomy, competence and relatedness), and, the students’ perception of a teacher’s more controlling style (judging and devaluing) will positively predict the psychological need thwarting. This, will positively predict amotivation, and this, will negatively predict disciplinary behaviours in the classroom and positively predict undiscipline behaviours”. (Line 321-326).
Line 336. I suggest that you should use “and” instead of “or”.
Response: We have modifed it.
Line 339. I believe “Hearens” is a typo. Also, please note that Haerens et al. (2015) examined need frustration, not need thwarting.
Response: We have made a mistake in concept. We have modified it (Line 341) and left the studio since we find it very interesting and related to our study.
Lines 336-351: I believe a study by Tilga et al. (2019) could provide a strong rationale here to highlight the importance of minimizing controlling behaviour, rather than an exclusive focus on enhancing autonomy-supportive behaviour, to enhance students’ adaptive outcomes. The reason is that students are sensitive to controlling behaviours even if they experience autonomy-supportive behaviour from their PE teachers.
Tilga, H., Hein, V., Koka, A., Hamilton, K., & Hagger, M. S. (2019). The role of teachers’ controlling behaviour in physical education on adolescents’ health-related quality of life: Test of a conditional process model*. Educational Psychology, 39(7), 862–880. https://doi.org/10.1080/01443410.2018.1546830
Lines 359-361: I believe a recent study by Koka et al. (2019) is very relevant here (see Figure 1 in a study by Koka and colleagues).
Koka, A., Tilga, H., Kalajas-Tilga, H., Hein, V., & Raudsepp, L. (2019). Perceived Controlling Behaviors of Physical Education Teachers and Objectively Measured Leisure-Time Physical Activity in Adolescents. International Journal of Environmental Research and Public Health, 16(15), 2709. https://doi.org/10.3390/ijerph16152709
Response: We welcome suggestions for references. We have introduced a paragraph (Lines 380-389) highlighting the importance of modifying the controlling styles by physical education teachers, including suggested bibliography, as well as a recent study by Koka et al. (2020):
Koka, A., Tilga, H., Kalajas-Tilga, H., Hein, V., & Raudsepp, L. (2020). Detrimental Effect of Perceived Controlling Behavior from Physical Education Teachers on Students’ Leisure-Time Physical Activity Intentions and Behavior: An Application of the Trans-Contextual Model. International Journal of Environmental Research and Public Health, 17(16), 5939.
https://doi.org/10.3390/ijerph17165939